# Cosmetic Formulation Based on an Açai Extract

**Roberta Censi [1], Dolores Vargas Peregrina [1], Giovanna Lacava [2], Dimitrios Agas [2], Giulio Lupidi [1], Maria Giovanna Sabbieti [2] and Piera Di Martino [1,*]**

[1]   School of Pharmacy, University of Camerino, 62032 Camerino, Italy; roberta.censi@unicam.it (R.C.);
     dolores.vargas@unicam.it (D.V.P.); giulio.lupidi@unicam.it (G.L.)
[2]   School of Biosciences and Veterinary Medicine, University of Camerino, 62032 Camerino, Italy;
     giovanna.lacava@unicam.it (G.L.); dimitrios.agas@unicam.it (D.A.); giovanna.sabbieti@unicam.it (M.G.S.)
[*]   Correspondence: piera.dimartino@unicam.it; Tel.: +39-320-7985643

**Abstract:** (1) Background: Açai berry extract is known for its high content in polyphenols and thus is a promising ingredient for cosmetic antiaging formulations; (2) Methods: In this study, the açai extract was firstly evaluated for its total phenol content (Folin Ciocalteau essay) and antioxidant activity (radical scavenging activity—DPPH; radical cation scavenging capacity—ABTS; ferric reducing antioxidant capacity—FRAP). Next, the açai extract was included in an O/W formulation and again was evaluated for its polyphenol content and antioxidant capacity. The formulation was tested for general characteristics, physicochemical properties and microbial stability. The proliferative effect on human immortalized fibroblasts was evaluated by the MTT essay, while TAC assay served to confirm that fibroblasts are protected from UV irradiation. The irritant potential was verified on 20 volunteers. The study concluded with the assessment of the sensorial characteristics of the cosmetic formulation; (3) Results: The pure açai extract exhibited high polyphenol content and antioxidant activity, and these characteristics were preserved in the O/W formulation as well. The O/W cosmetic formulation proved to be stable under accelerated and normal conditions, and the preservatives were successful in challenging the resistance against microbial contamination. The mean irritant potential was zero in all volunteers, and the cosmetic formulation showed a good sensorial profile; (4) Conclusions: Açai extract is an interesting ingredient for cosmetic antiaging formulations.

**Keywords:** açai berry extract; anti-aging; antioxidant; O/W emulsion

## 1. Introduction

The açai berry is the fruit of a palm tree (*Euterpe oleracea*) from the Amazon forest in Brazil. It is largely consumed in Brazil particularly as food, but because of its properties, interest in it has expanded worldwide [1].

Considerable interest has focused on the açai's high antioxidant capacity, which has attributed to its polyphenolic compounds [2,3], over 90% of which are anthocyanins [4]. Anthocyanins seem to have a photoprotective effect by directly eliminating reactive oxygen species during photooxidative stress [5]. High amounts of procyanidin trimers and dimers, (+)-catechin, vanillic acid, and syringic acid have been identified in açai fruit [6]. Other non-anthocyanin polyphenol and phenolic acids are also present [2,4,7,8]. Taking into account the antioxidant and photoprotective properties of açai fruit, the objective of this study was the formulation of an O/W emulsion with UV protective and antiaging properties. The objective was to formulate an emulsion containing an açai berry extract with good sensorial characteristics and appropriate properties for cosmetic use. The emulsion was thus evaluated for its physicochemical and microbial stability, antioxidant properties, and sensorial characteristics.

## 2. Materials and Methods

### 2.1. Materials

The dry açai extract (AE) (*Euterpe Oleracea*) was kindly supplied by A.C.E.F. (Fiorenzuola D'Arda, Italy). 1,1-Diphenyl-2-picrylhydrazyl (DPPH), 2,4,6-Tris(2-pyridyl)-s-triazine (TPTZ), (±)-6-Hydroxy-2,5,7,8-tetramethylchromane-2-carboxylic acid (TROLOX), 2,2′-Azino-bis(3-ethylbenzothiazoline-6-sulfonic acid) diammonium salt (98% TLC) (ABTS, gallic acid, sodium carbonate monohydrate ACS reagent, sodium acetate and ethanol, ethanol absolute grade) were purchased from Sigma-Aldrich (Stenheim, Germany). Manganese (IV) oxidize activated ($\geq$90%) and Folin Ciocalteu's phenol reagent were purchased from Fluka (Buchs, Switzerland). Anhydrous sodium acetate and Iron(III) anhydrous hydrochloride were purchased from J.T. Baker Analyzed (Center Valley, PA, USA) and sodium carbonate anhydrous was purchased from Carlo Erba (Milano, Italy). All solvents and reagents were of analytical grade.

Ultrapure water was produced by Gradient Milli-Q® (Millipore, Molsheim, France).

For the cosmetic formulation, the following ingredients were used, kindly provided by the respective suppliers: Emulium Kappa® 2 (Inci: Candelilla, jojoba, ridebranpolyglycerul-3 esters, glyceryl stearate) (Gattefossé, Lyon, France); butyrospermum parkii butter, Avicel® PC 591 (Inci: microcrystalline cellulose, cellulose gum) (Biochim, Casarile, MI, Italy); Cetiol® LC (Inci: cococaprylate/caprate) (Basf, Ludwigshafen am Rhein, Germany); Cetyl stearyl alcohol, Glycine soja oil, citric acid (ACEF, Fiorenzuola D'Arda, Italy); Euxyl® K712 (Inci: sodium benzoate, potassium sorbate, water) (Schülke & Mayr Italia, Milan, Italy); Natralquest® E30 (Inci: Trisodium Ethylenediamine Disuccinate) (Innospec, Littleton, CO, USA). The exact formulation of the O/W emulsion is given in Table 1.

**Table 1.** Formulation of the O/W emulsion based on açai extract.

| Phase | Commercial Name and Supplier | INCI Name | Amount (%) | Function |
|---|---|---|---|---|
| Phase A Oil Phase | Emulium kappa 2 | Candelilla jojoba, ridebranpolyglycerul-3 esters, glyceryl stearate | 6.0 | Emulsifier |
| | Alcool cetilstearilico TA 1618 | Cetearyl alcohol | 2.0 | Thickning agent |
| | Karité butter | Butyrospermum Parkii butter | 2.0 | Emollient |
| | Cetiol LC | Coco Caprilate/Caprate | 5.0 | Emollient |
| | Soja oil | Glycine Soja Oil | 5.0 | Emollient |
| Phase B Acqueous phase | Water | Aqua | q.s. 100.0 | Solvent |
| | Avicel PC 591 | Microcrystalline cellulose (and) cellulose gum | 1.5 | Thickening agent |
| Phase C | Titrated açai extract | Acai Extract | 0.5, 1.0, 2.0 | Functional |
| Phase D | Euxyl K712 | sodium benzoate, potassium sorbate, water | 1.0 | Preservative |
| | Natrlquest E30 | Trisodium Ethylenediamine Disuccinate | 0.5 | Chelant |
| | Citric acid | Citric acid | q.s. to pH 5.0 | Acidifier |

### 2.2. Total Phenol Content Determination and Evaluation of Antioxidant Capacity

The Total Phenol Content (TPC) of açai extract and O/W formulation was determined according to the Folin-Ciocalteu spectrophotometric method [9] with some modifications [10]. Briefly, the açai extract was dissolved in water (1 mg/mL *w/w*) and 50 µL of this solution was added to 150 µL of Folin-Ciocalteu's phenol reagent, diluted 1:4 with water. Then, 50 µL of Na$_2$CO$_3$ saturated solution were added. After incubation at room temperature for 10 min, the absorbance of each well was determined at 765 nm using a microplate reader (FLUOstar Omega, BMG Labtech GmbH, Ortenberg, Germany). The measurement was compared to a calibration standard solution of gallic acid (GA), and results were expressed as micrograms of gallic acid equivalents (GAE) per grams of sample (mg GAE/g).

The antioxidant activity was evaluated by measuring 1,1-diphenyl-2-picrylhydrazyl (DPPH$^\bullet$) radical scavenging activity, 2,2'-azino-bis(3-ethylbenzothiazoline-6-sulphonic acid) (ABTS$^{\bullet+}$) radical cation scavenging capacity, and Ferric Reducing Antioxidant Capacity (FRAP).

Trolox (6-hydroxy-2,5,7,8-tetramethylchroman-2-carboxylic acid) was used as the calibration standard. Values were expressed as IC$_{50}$, defined as the concentration of the tested material required to cause a 50% decrease in initial DPPH, ABTS or iron concentration, as well as μmol Trolox equivalent/g of sample.

DPPH free radical scavenging activity was evaluated on a microplate analytical assay according to the previously published methods [11] with some modifications [12]. Briefly, the açai extract was dissolved in water (1 mg/mL $w/w$) and 50 μL aliquot of the sample (concentration of 10 mg mL$^{-1}$) and standard were added to 150 μL of DPPH in absolute ethanol in a 96-well microtitre plate (BD Falcon$^{\text{TM}}$). After incubation at 37 °C for 20 min, the absorbance of each well was determined at 517 nm using a microplate reader.

The ABTS assay was performed following the procedures [13] applied to a 96-well microliter plate assay [12]. The ABTS$^{\bullet+}$ solution (5 mM) was prepared by oxidizing ABTS with MnO$_2$ in water for 30 min in the dark. A 50 μL aliquot of the different concentrations of sample and standard (trolox) were added to 150 μL of ABTS$^{\bullet+}$ solution in a 96-well microtitre plate (BD Falcon$^{\text{TM}}$). After incubation at room temperature for 10 min, the absorbance of each well was determined at 734 nm using a microplate reader. The FRAP values were determined according to a previously published method [14] with some modifications [15].

The FRAP reagent was prepared by mixing the following three solutions:

(1) 50 mL 0.3 M acetate buffer pH 3.6 (1.23 g of sodium acetate in 50 mL of water acidifying with acetic acid);
(2) 5 mL of stock solution of 5 mM TPTZ (2,4,6-tripyridyl-s-triazine) (15.6 mg) in 40 mM HCl;
(3) 5 mL of 5 mM FeCl$_3$·6 H$_2$O (16.2 mg) in 40 mM HCl.

The FRAP reagent was heated at 37 °C before use. Aliquots of a 25 μL sample (solutions at the concentration of 10 mg mL$^{-1}$) were added in triplicate onto wells of a 96-well plate (BD Falcon$^{\text{TM}}$). The assay was started by adding 175 μL of FRAP reagent to each well. The plate was immediately shaken in a FLUOstar Omega plate reader for 30 s and the reaction was allowed to run for 10 min after which the plate was read on a plate reader (593 nm). A reference solution of Trolox was run simultaneously and used to generate the calibration curve by linear regression. The standard curve was linear between 25 and 800 μM Trolox (TE). Results were expressed in μM trolox equivalent (TE) g$^{-1}$ sample.

### 2.3. Preparation of the O/W Emulsion

The ingredients were carefully weighted and four different phases were prepared (Table 1). The two phases, A and B (oil and water), were heated at 70 °C and then the oil phase was added to the aqueous phase under vigorous and constant stirring. Mechanical stirring was continued by a homogenizer (Ultraturrax IKA$^\circledR$ T25, Staufen, Germany) at the speed rate of 3000 rpm while reaching a temperature of 40 °C. Under cooling, the emulsion increased in viscosity and homogeneity. When the emulsion reached a temperature of 30 °C, the phases C and D were consecutively added under gentle stirring. Four O/W formulations were prepared: one without açai extract, and three with 0.5, 1.0, and 2.0% of açai extract.

### 2.4. Physicochemical Characterization of O/W Cream

The O/W emulsion was analysed for pH with a Jenway 3520 pH meter (Staffordshire, UK), equipped with a Jenway 924007 electrode specific for cosmetic applications. The O/W emulsion was analysed for density by measuring the volume in a glass cylinder of a carefully weighed amount of emulsion.

The dispersion of the oil in water emulsion was evaluated under an optical microscope (Meiji Techno Co., San José, CA, USA) at a magnification of 100×.

Colour space measurements were performed using a Minolta CR-400 model spectrophotometer (Minolta Camera Co., Osaka, Japan) according to the method recommended by the International Commission on Illumination (CIELab 1978). Calibration was conducted on a white plate (Y = 86.8, X = 0.3167, Z = 0.3237) to standardize the equipment before each colour measurement. Samples were transferred onto a glass plate and several measurements were performed.

The accelerated physicochemical stability of the O/W emulsion was evaluated by centrifuging an appropriate sample amount (Micro-centrifuge Scilogex D3024R, Rocky Hill, CT, USA) at 4000 rpm at 20 °C for 30 min. The evaluation of the stability under accelerated conditions was also performed by temperature shock, by placing the formulation alternatively at 4 and 40 °C three times. Reference samples were meanwhile tested at room conditions for 1 year (long term stability).

## 2.5. Challenge Test

The challenge test was performed according to the Method ISO 11930:2012 Cosmetics, Microbiology, Evaluation of antimicrobial protection of a cosmetic product. The antimicrobial activity was screened against *Pseudomonas aeruginosa* ATCC 9027, *Staphylococcus aureus* ATCC 6538, *Candida albicans* ATCC 10231, *Aspergillus brasiliensis* ATCC 16404 and *Escherichia coli* ATCC 87394. *Escherichia coli, Staphylococcus aureus,* and *Pseudomonas aeruginosa* were incubated for 48–72 h at 30 °C on Tryptic Soy Agar (TSA) (Sigma Aldrich, Stenheim, Germany), *Candida albicans* was incubated for 48–72 h at 30 °C on a Sabouraud Dextrose Agar (SDA) (Sigma Aldrich, Stenheim, Germany), and *Aspergillus brasiliensis* was incubated for 72–120 h at 22.5 °C on a potato Dextrose Agar (PDA) (Sigma Aldrich, Stenheim, Germany). Initial inoculi were 700,000, 480,000, 300,000, 41,000, and 9000 UFC/g for *E. coli, S. aureus, P. aeruginosa, C. albicans, A. brasiliensis,* respectively.

The test was followed for 28 days.

## 2.6. In Vitro Experiments

For in vitro tests (MTS assay, Western Blot and TAC assay), BJ-5ta human fibroblast cell lines (ATTCC, LGC Standards S.r.L., Milano, Italy) were used. Cells were grown in Dulbecco's modified Eagle medium/MED 199 MATASSE 3:1 (DMEM/MED 199 MATASSE) (Life Technologies, Milano, Italy), to which were added 10% of the heat inactivated foetal calf serum (HI-FCS, Life Technologies, Milano, Italy) penicillin and streptomycin.

## 2.7. Assessment of the Metabolic Activity of Viable Cells (MTS)

The metabolic activity of viable cells was determined by the MTS [3-(4,5-dimethylthiazol-2-yl)-5 (carboxymethoxyphenyl)-2-(4-sulfophenyl)-2*H*-tetrazolium] assay. One part of the dry extracts and the O/W emulsions was used for MTS Assay. Bj-5TA human fibroblasts were plated at the density of 5000 cells/well in 96 culture dishes and were grown until 80% of confluence. Cultures were treated for 24 h with the above-described compounds at different concentrations. Control cultures were treated with the appropriate vehicle. Then, cells were incubated with 20 μL/well of CellTiter 96 Aqueous One Solution Reagent (Promega Italia, Milano, Italy) for 2 h in a humidified, 5% $CO_2$ atmosphere. The quantity of formazan product is directly proportional to the number of living cells in the culture. The colored formazan was measured by reading the absorbance at 490 nm using a 96-well plate reader.

## 2.8. Total Antioxidant Capacity (TAC)

The total antioxidant capacity of the dry extract and the O/W emulsions was determined by the Total Antioxidant Capacity (TAC) Colorimetric Assay Kit (Biovision, Milpitas, CA, USA).

BJ-5TA fibroblasts were plated at a density of 5000 cells/well in 96 culture dishes and grown until 80% confluence. Cultures were treated with the above-described compounds at different concentrations or with vehicle, and were immediately exposed to UV irradiation for 4 min in order to stimulate ROS

production. Control cultures were divided into two subgroups: One subgroup was exposed to UV irradiation, while the other was not irradiated. Cultures were maintained at 37 °C in a humidified, 5% $CO_2$ atmosphere. After 24 h, the antioxidant activity of the compounds was evaluated using the TAC Colorimetric Assay Kit, according to the manufacturer's instructions. Briefly, the culture's supernatants from all experimental groups were incubated with $Cu^{2+}$ working solution at RT for 1.5 h. The total antioxidant capacity was then measured by reading the absorbance at 570 nm using a 96-well plate reader.

### 2.9. Local Compatibility Test with Human Skin (Irritant Potential)

This test was performed to evaluate the compatibility of the cosmetic formulation with human skin (irritant potential), under normal condition of use, in compliance with the Helsinki Declaration (64th WMA General Assembly, Fortaleza, Brazil, October 2013) and according to COLIPA guidelines [16]. The substance was left in contact with the skin for 48 h (model Curatest® F, adhesive strips for patch test, Lohmann and Rauscher International, Rengsdorf, Germany) in a sufficient amount to fill a 1 cm² test disk (approximately 0.07–0.1 mL). The assessment is made by comparison with a negative control. The skin reactions were evaluated 15 min after patch removal, and again after 24 h, according to defined parameters (erythema, desquamation, oedema, and vescicles). The test was performed in single blind mode, under the direction of a medical doctor certified in dermatology.

### 2.10. Açai Formulations Sensory Analysis

A sensory analysis was conducted according to ISO 11136:2014, using a panel of 20 volunteers (average age $25.5 \pm 2.0$ years old) of female gender, without dermatological diseases, recruited based on their interest and availability to participate, and who enrolled after giving their informed consent. The panel provided a sensory analysis of the different formulations. Each volunteer answered a number of questions about the sensory characteristics of the formulations, such as texture (consistency and stickiness), skin feel during application (ease of application, spreadability, oiliness) and skin feeling after application (hydration, softness of the skin and freshness). Responses were given on a scale from 1 to 5. Sensory parameters were evaluated by applying a small amount of each formulation onto the fingertips and then rubbing the formulation into the skin. A radar graph summarizes the results.

### 2.11. Statistical Analysis

All tests were conducted in triplicate. Data are reported as means $\pm$ SD. Analysis of variance and significant differences among means were tested by one-way ANOVA using GradPad Prism 7 computer program (GradPad Software, San Diego, CA, USA).

## 3. Results and Discussion

### 3.1. General Characteristics of the Açai Formulation

The general properties of the O/W emulsions (that without açai extract, and those containing 0.5, 1.0 and 2.0% of açai extract) are given in Table 2. pH ranged between 5.02 and 5.05 after acidification with citric acid. This pH range is necessary to respect the skin pH and because the preservatives (sodium benzoate, potassium sorbate) are effective at those levels. Density was similar among the different formulations, being approximately $0.98 \pm 0.03$ g/L.

Colour space measurements (Table 2) were expressed as the two colour coordinates a* and b*. a* takes positive values for a reddish colour and negative ones for a greenish colour, while b* takes positive values for a yellowish colour and negative ones for a bluish colour. The emulsion without the açai extract is characterized by negative a* and b*, which means there is a tendency to greenish and bluish colours. These values turn into positive ones in samples containing the açai extract. a* values increased progressively when increasing the açai extract content, meaning that the sample was enriched in the reddish components, while b* progressively increased, meaning that when the açai

extract content in the sample increased, the sample had a more yellowish colour. The psychometric index of lightness L* is an estimation of the relative luminosity, and any given colour can be regarded as equivalent to a grey scale, between black (L* = 0) and white (L* = 100). The L* value for the sample without açai extract is rather high, indicating a value closer to the white component. This value progressively decreased with increases in açai extract content. A picture of the emulsions is presented in Figure 1.

The formulation was stable under accelerated conditions and in the long term. No instability phenomena, such as separation phase or change in color or odor, were observed.

The challenge test demonstrated the efficacy of the preservative system chosen for the formulation. The formulation was highly tolerated by normal human skin, as shown by the Mean Irritation Index of 0 observed 15 min after application, and again 24 h after application for all of the tested volunteers.

**Table 2.** General properties of açai formulations.

| Parameters | | O/W Emulsion without Açai Extract | O/W Emulsion with 0.5% Açai Extract | O/W Emulsion with 1% Açai Extract | O/W Emulsion with 2% Açai Extract |
|---|---|---|---|---|---|
| pH | | 5.03 ± 0.12 | 5.02 ± 0.14 | 5.04 ± 0.08 | 5.05 ± 0.11 |
| Density (g/L) | | 0.98 ± 0.02 | 0.98 ± 0.01 | 0.98 ± 0.02 | 0.98 ± 0.03 |
| Color | L* | 75.21 | 66.98 | 63.80 | 58.46 |
| | a* | −1.23 | 5.49 | 7.84 | 11.30 |
| | b* | −2.98 | 1.75 | 3.27 | 4.51 |
| Centrifugation | | No separation phase | No separation phase | No separation phase | No separation phase |
| Accelerated stability | | Stable | Stable | Stable | Stable |
| Long term stability | | Stable | Stable | Stable | Stable |
| Challenge test | | Passed (Criterion A) | Passed (Criterion A) | Passed (Criterion A) | Passed (Criterion A) |
| Mean Irritation Index (after 15 min) | | 0 | 0 | 0 | 0 |
| Mean Irritation Index (after 24 h) | | 0 | 0 | 0 | 0 |

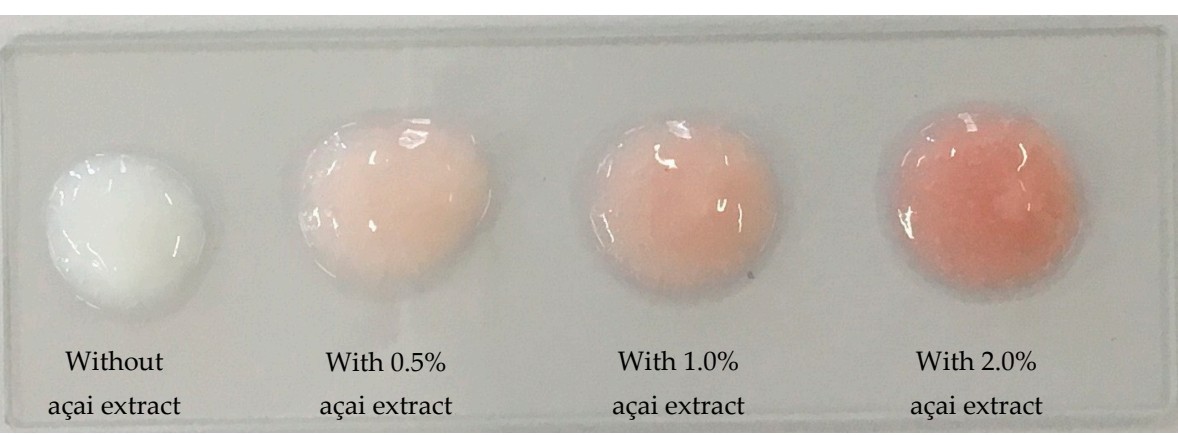

| Without açai extract | With 0.5% açai extract | With 1.0% açai extract | With 2.0% açai extract |

**Figure 1.** Colours of the emulsions according the presence of the açai extract.

A prediction of the physicochemical stability of the formulation could be obtained by the observation of the emulsion under optical microscopy. The distribution of the droplets of the oil phase in the water phase is presented in Figure 2. From this Figure, the homogeneous distribution of the droplets of the internal phase (oil phase) into the external one (water phase) is evident. In addition, droplets show a homogenous size distribution. No double emulsion was formed.

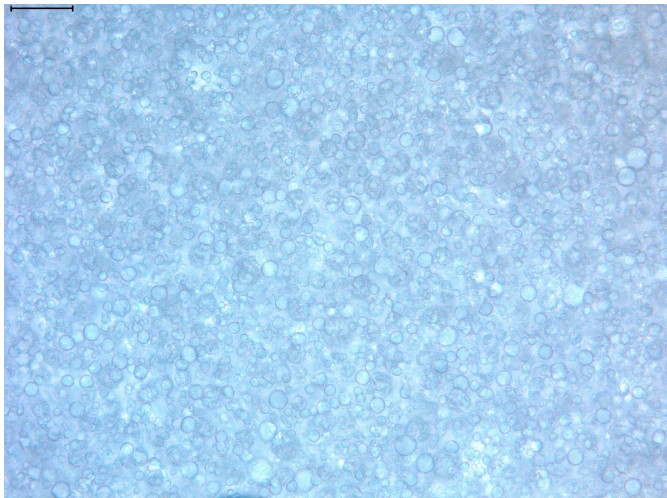

**Figure 2.** Droplets of the internal oil phase in the O/W emulsion. Bar corresponds to 10 μm. Magnification 100×.

The rheological behaviour of the studied formulations was investigated by thixotropic loop experiments, as shown in Figure 3. No significant differences in the rheological behaviour of the creams containing increasing concentration of açai extract were found. All the formulations showed a shear-thinning and thixotropic behaviour, with a marked tendency to decrease their viscosity values, at increasing shear rates, and no recovery of the initial viscosity values when the shear rate was removed. High viscosity at rest (or at low stresses) usually slows down effects like phase separation and, therefore, improves the shelf life of a formulation. Low viscosity in the medium-to-high shear rate range is preferred for easy application of the cosmetic cream on the skin. Delayed recovery improves the absorption of the cream through the skin.

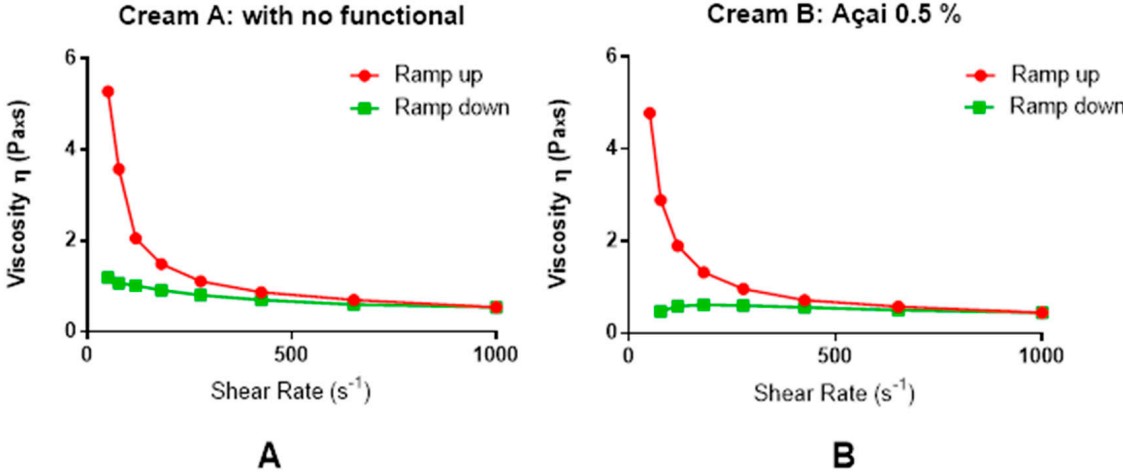

**Figure 3.** *Cont.*

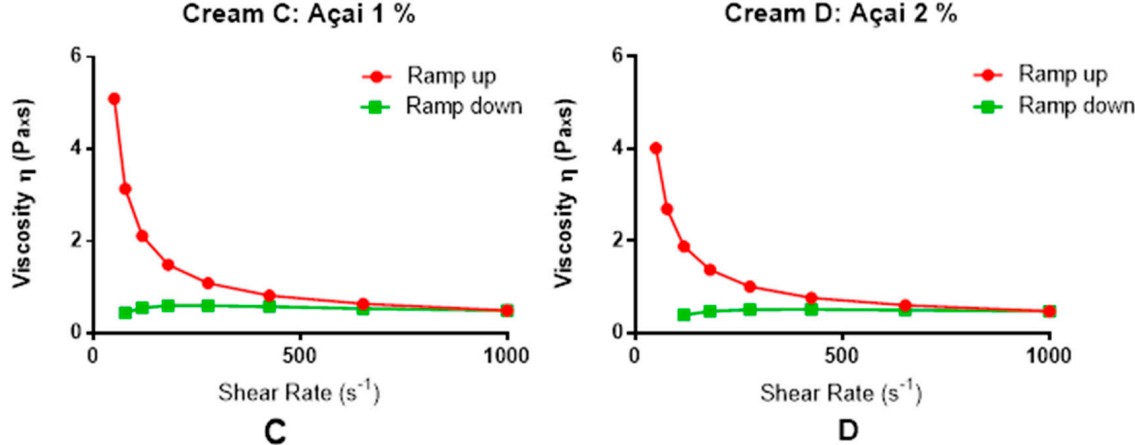

**Figure 3.** Results of thixotropic loop experiments for cream formulations at increasing açai extract. (**A**) Cream with 0% açai extract; (**B**) cream with 0.5% açai extract; (**C**) ream with 1% açai extract; (**D**) cream with 2% açai extract.

### 3.2. Total Phenol Content and Antioxidant Activity of O/W Emulsions

Results of the antioxidant capacity of the açai extract and formulations with açai extract at different percentages are reported in Table 3. The pure açai extract shows values very similar to those reported by the literature [17]. These values appeared high if compared to other extracts known for their total phenol content and antioxidant capacity [18]. A considerable decrease in the total phenol content and antioxidant capacity was observed in the emulsions due to the dilution of the extract in the emulsions.

**Table 3.** Results of total phenol content and antioxidant activity for açai extract and formulations containing different percentages of açai extract.

| Product | FOLIN CIOCALTEAU (µg GAE/g) | DPPH (µmoli TE/g) | ABTS (µmoli TE/g) | FRAP (µmoli TE/g) |
|---|---|---|---|---|
| Açai extract | $35.50 \pm 2.6$ | $148.4 \pm 10.9$ | $57.8 \pm 5.2$ | $134.8 \pm 8.2$ |
| O/W Emulsion without açai extract | $2.3 \pm 0.2$ | $5.8 \pm 1.2$ | $0.70 \pm 0.3$ | $1.2 \pm 0.1$ |
| O/W Emulsion with 0.5% açai extract | $20.\,3 \pm 0.8$ | $66.8 \pm 1.5$ | $21.02 \pm 0.5$ | $52.8 \pm 1.2$ |
| O/W Emulsion with 1.0% açai extract | $25.8 \pm 0.7$ | $70.6 \pm 1.8$ | $20.01 \pm 0.5$ | $55.2 \pm 1.2$ |
| O/W Emulsion with 2.0% açai extract | $29.2 \pm 0.7$ | $82.5 \pm 2.5$ | $13.30 \pm 0.4$ | $60.7 \pm 1.5$ |

### 3.3. Effects of Açai on Metabolic Activity of Viable BJ-5TA Human Fibroblasts

To determine the effects of açai alone or emulsified on the metabolic activity of viable BJ-5TA cells, MTS assays were carried out. Treatment with açai emulsion from 1% to 2.0% for 24 h caused a statistically significant increase of cell viability. The maximum increase was reached by the treatment with 100% açai (Figure 4A).

### 3.4. Antioxidant Activity of Açai on BJ-5TA Human Fibroblasts

The antioxidant activity of the açai alone or emulsified was evaluated by TAC Assay. As detailed in the Materials and Methods section, BJ-5TA fibroblasts were exposed for 5 min to UV irradiation after treatment.

As shown in Figure 4B, no statistically significant effects were found in terms of antioxidant activity of the açai alone compared with the control (UV exposed cells). Interestingly, antioxidant effects were found not only in cultures treated with the 2% açai emulsion but also in those treated with the 1% emulsion.

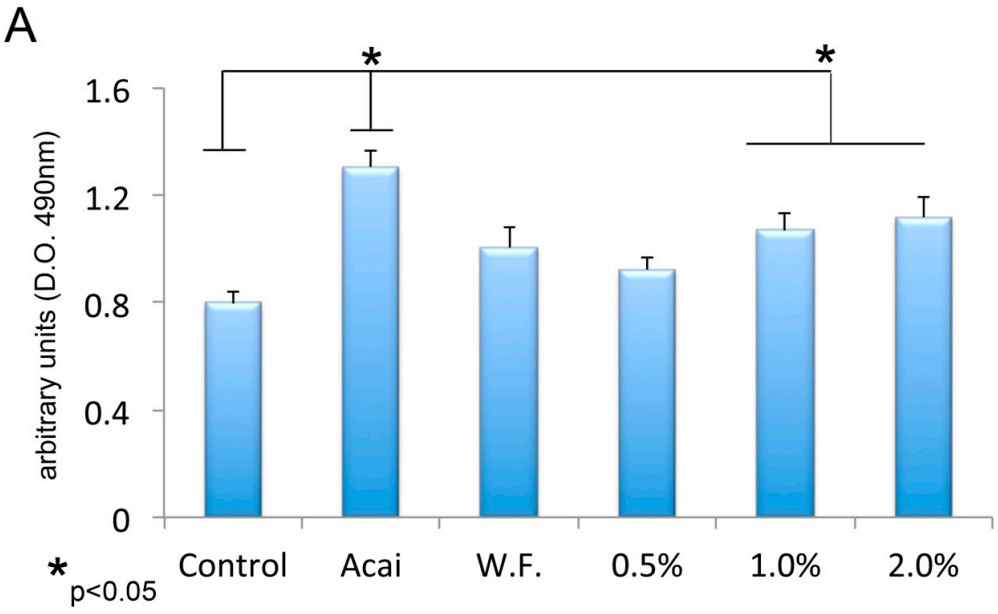

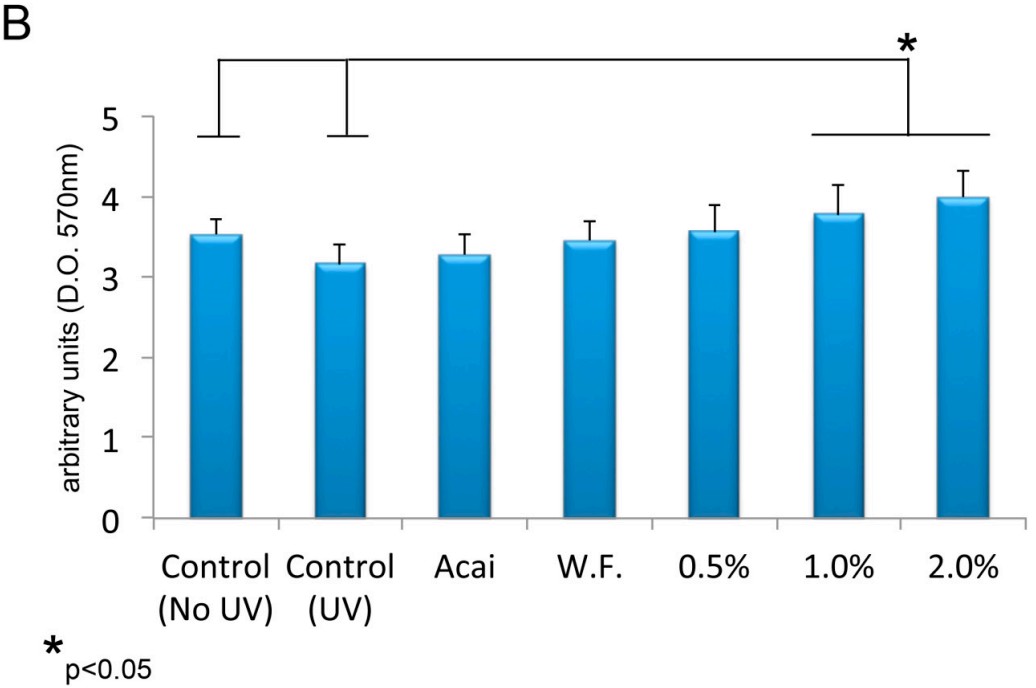

**Figure 4.** Results for the assessment of the metabolic activity of viable cells (MTS) (**A**) and Total Antioxidant Capacity (TAC); (**B**) for pure açai extract (Acai), formulations without açai extract (W.F.), formulations with 0.5, 1, and 2% of açai extract. For the TAC essay, experiments were carried out in presence of two controls, one corresponding to non-irradiated cells, and one corresponding to irradiated cells.

*3.5. Sensory Analysis*

Due to the good stability of the formulation containing 2.0% of açai extract, considering that it showed the best in vitro results, and also taking into account that a 2.0% concentration is appropriate for human use and did not cause any adverse effects on normal skin, this formulation was selected for the sensory analysis, the results of which are indicated in Figure 5. The formulation containing 2.0% of açai

extract proved to be very pleasant for cosmetic use. Actually, freshness, spreadability, and application, all parameters affecting the pleasantness of a cosmetic formulation for female volunteers of an average age $25.5 \pm 2.0$ years old, were classified at their highest rate, while oiliness, that is in general not a favorable factor for a cosmetic cream, was classified at the lowest rate. Consistency was classified at an intermediate value, while stickiness, that is unpleasant, was classified at a very low value. The cosmetic performances expressed as hydration and skin softness were also classified at their highest values.

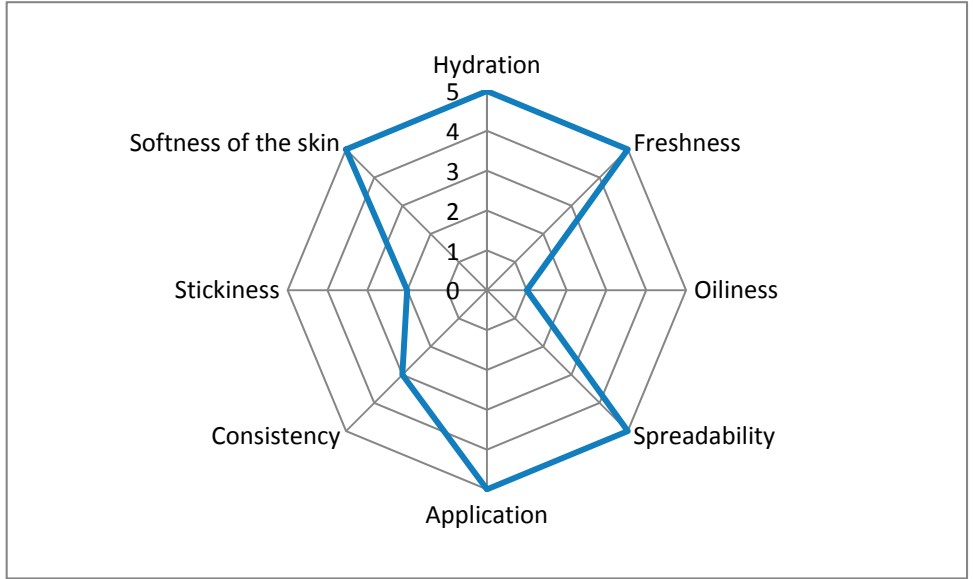

**Figure 5.** Results of sensory analysis of the emulsion with 2.0% of açai extract.

## 4. Conclusions

The present study demonstrated that an O/W formulation containing 2.0% of an açai extract not only displays cosmetic efficacy as UV protection and possesses anti-aging properties, but also has pleasant sensorial characteristics for cosmetic use.

**Author Contributions:** P.D.M. and R.C. conceived and designed the experiments; G.L. was responsible for the evaluation of total phenol content and antioxidant capacity; G.L. and M.G.S. were responsible for the in vitro experiments; D.A. was responsible for statistical analysis; D.V.P. was responsible for cosmetic formulation and characterization; R.C. was also responsible for testing in humans. All authors contributed to the manuscript preparation.

**Funding:** This research was funded by European Commission: H2020-MSCA-ITN-2015 ISPIC project (grant number 675743); H2020-MSCA-RISE-2016 CHARMED project (grant number 734684); H2020-MSCA-RISE-2017 CANCER project (grant number 777682).

**Acknowledgments:** The authors would like to thank Gaia Bartolini for her contribution to the experimental work and Sheila Beatty for editing the English usage of the manuscript.

**Conflicts of Interest:** The authors declare no conflict of interest.

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
