# Peer review of "Cosmetic Formulation Based on an Açai Extract"

_cosmetics, doi:10.3390/cosmetics5030048_

Round 1

Reviewer 1 Report

In the paper entitled ‘Antiaging cosmetic formulation based on an Açai extract’ the açai extract was evaluated for its total phenol content and antioxidant activity and then it was included in an O/W formulation and again evaluated for polyphenol content and antioxidant capacity. The obtained emulsion was tested including physicochemical properties, microbial stability and sensorial characteristics.  Euterpe Oleracea (Acai) Fruit Extract is used in cosmetics as antioxidant, anti-aging and anti-inflammatory agent and it can also be found in cosmetic emulsions. Therefore, the conclusions from this publication are not new. Please clearly indicate the element of novelty in this paper. The review of the research on acai fruit extract for cosmetic applications should also be considered in the part “introduction” of manuscript. Moreover, during sensory analysis the stickiness and consistency of emulsion with 2% Açai extract were not well assessed, while the Authors suggest, that this formulation proved to be very pleasant for cosmetic use. Please explain. Please also provide more information about the 12 volunteers, including their age and gender.

Author Response

R: First of all, Authors acknowledge the reviewer for her/his kind support in revising the manuscript.

Our comments are at the end of reviewer’s comments.

In the paper entitled ‘Antiaging cosmetic formulation based on an Açai extract’ the açai extract was evaluated for its total phenol content and antioxidant activity and then it was included in an O/W formulation and again evaluated for polyphenol content and antioxidant capacity. The obtained emulsion was tested including physicochemical properties, microbial stability and sensorial characteristics.  Euterpe Oleracea (Acai) Fruit Extract is used in cosmetics as antioxidant, anti-aging and anti-inflammatory agent and it can also be found in cosmetic emulsions.

Therefore, the conclusions from this publication are not new. Please clearly indicate the element of novelty in this paper. The review of the research on acai fruit extract for cosmetic applications should also be considered in the part “introduction” of manuscript.

R: Actually there are several patents, but it was really difficult to find a scientific manuscript reporting a systemic study of the development of a cosmetic formulation containing an açai extract.

Moreover, during sensory analysis the stickiness and consistency of emulsion with 2% Açai extract were not well assessed, while the Authors suggest, that this formulation proved to be very pleasant for cosmetic use. Please explain.

R: An explanation has been added.

Please also provide more information about the 12 volunteers, including their age and gender.

R: More information was provided.

Reviewer 2 Report

The Authors of this manuscript undertook an interesting studies regarding possible application in cosmetics acai berries, well known between dieticians. Application of açai extract in creams was successful. Due to radical scavenging properties acai berries can be used especially in antiaging cosmetics. The studies deliver information about action of açai extract on the skin after topical application. The results are very promising. It seems that the extracts could be applied to produce novel antiaging creams.

The manuscript is ready for publication.

Author Response

The Authors of this manuscript undertook an interesting studies regarding possible application in cosmetics acai berries, well known between dieticians. Application of açai extract in creams was successful. Due to radical scavenging properties acai berries can be used especially in antiaging cosmetics. The studies deliver information about action of açai extract on the skin after topical application. The results are very promising. It seems that the extracts could be applied to produce novel antiaging creams.

The manuscript is ready for publication.

R: Authors acknowledge the reviewer for her/his kind support in revising the manuscript.

Reviewer 3 Report

The manuscript entitled “Antiaging cosmetic formulation based on an Açai extract” by Censi et al. evaluates not only the açai extract but also a formulation without and with different percentages of this extract as active ingredient. The manuscript is in the journal Scopus.  Nevertheless, the English clearly needs to be improved as well as gramma. In my opinion the paper is interesting but it is not in conditions yet to be accept. My major concerns:

-          The title does not reflect the article results. The paper is about a possible cosmetic formulation with açai and any anti-aging assays were performed. The title should be revised.

-          For the formulation TPC determination, which solvent was used to dissolve the emulsion? I do not believe that water is enough.

-          The pH was directly measured on the pH meter?

-          Why for the accelerated physicochemical stability the authors used 30 °C instead of 40 °C , according to the international guidelines? Was the relative humidity controlled?

-          The authors should include the ethical approval number on the in vivo assay.

-          Which was the negative control used for the in vivo assay?

-          In my opinion 12 volunteers are not enough to make statistical analyses for the sensorial prove.

-          Why the cells assays were not expressed as cell viability (%) vs concentration?

Author Response

R: First of all, Authors acknowledge the reviewer for her/his kind support in revising the manuscript.

Our comments are at the end of reviewer’s comments.

Comments and Suggestions for Authors

The manuscript entitled “Antiaging cosmetic formulation based on an Açai extract” by Censi et al. evaluates not only the açai extract but also a formulation without and with different percentages of this extract as active ingredient. The manuscript is in the journal Scopus.

Nevertheless, the English clearly needs to be improved as well as gramma. In my opinion the paper is interesting but it is not in conditions yet to be accept.

R: English has been reviewed by Mrs Sheila Betty, who is our English speaker lecturer.

My major concerns:

-          The title does not reflect the article results. The paper is about a possible cosmetic formulation with açai and any anti-aging assays were performed. The title should be revised.

R: The Title has been changed in: “Cosmetic formulation based on an Açai extract”

-          For the formulation TPC determination, which solvent was used to dissolve the emulsion? I do not believe that water is enough.

R: The TPC was carried out in water. The açai extract used for this study and provided by the supplier was already in water and preliminary essays demonstrated water was able to extract the antioxidant molecules.

-          The pH was directly measured on the pH meter?

R: Yes because the electrode is appropriate for measurements of cosmetic formulations such as O/W emulsions.

-          Why for the accelerated physicochemical stability the authors used 30 °C instead of 40 °C , according to the international guidelines? Was the relative humidity controlled?

R: Reviewer is right! Actually “30 °C” was indicated by error, and “40 °C” was the correct temperature used!

-          The authors should include the ethical approval number on the in vivo assay.

R: The cosmetic regulation does not request for the ethical approval for this kind of evaluations. The cosmetic is safe by definition and follows different Regulations than for example that of Pharma. A sentence has been added.

-          Which was the negative control used for the in vivo assay?

R: No negative control was used for this study as was the same for similar studies.

-          In my opinion 12 volunteers are not enough to make statistical analyses for the sensorial prove.

R: For in vivo study and the number of volunteers we referred to similar studies.

https://doi.org/10.1111/ics.12302

https://doi.org/10.1108/13612020310484799

Cosmetics 2018, 5, 26; doi:10.3390/cosmetics5020026

-          Why the cells assays were not expressed as cell viability (%) vs concentration?

R: The average of the raw data presented in the graph represents a more accurate  indication of the effects of the tested compounds.

Round 2

Reviewer 1 Report

The paper has improved since I last reviewed it. I recommend this paper for publication.

Author Response

We would like to thank the reviewer for her/his kind support.

Reviewer 3 Report

I still do not agree with some authors options, such as the evaluation of formulations TPC and DPPH using water to dissolve (in my opinion some constituints do not dissolve). Also, 12 volunteers are really few for the in vivo assays. The same opinion regarding the absence of a negative control for the in vivo assays.

Author Response

First of all, we would like to thank the reviewer for her/his kind support.

1) The extract was really completely dissolved in water (the solution was limpid) and antioxidant molecules are soluble in water. We always used this procedure for these test. 

2) We recovered additional evaluations from additional 8 volunteers. The graph did not changed and we confirm previous results.

3) In most studies no negative control was used:

International Journal of Cosmetic Science, 2014, 36, 159–166

Chemometrics and Intelligent Laboratory Systems 124 (2013) 21–31